# 2D-DIGE-MS Proteomics Approaches for Identification of Gelsolin and Peroxiredoxin 4 with Lymph Node Metastasis in Colorectal Cancer

**DOI:** 10.3390/cancers14133189

**Published:** 2022-06-29

**Authors:** Cheng-Yi Huang, Ko-Chao Lee, Shui-Yi Tung, Wen-Shin Huang, Chih-Chuan Teng, Kam-Fai Lee, Meng-Chiao Hsieh, Hsing-Chun Kuo

**Affiliations:** 1Division of Colon and Rectal Surgery, Department of Surgery, Chang Gung Memorial Hospital, Chiayi 61363, Taiwan; bluesky@cgmh.org.tw (C.-Y.H.); wen1204@cgmh.org.tw (W.-S.H.); 2Chang Gung Memorial Hospital—Kaohsiung Medical Center, Division of Colorectal Surgery, Department of Surgery, Chang Gung University College of Medicine, Kaohsiung 83301, Taiwan; kclee@cgmh.org.tw; 3Department of Hepato-Gastroenterology, Chang Gung Memorial Hospital, Chiayi 61363, Taiwan; ma1898@adm.cgmh.org.tw; 4College of Medicine, Chang Gung University, Taoyuan 33302, Taiwan; 5Division of Basic Medical Sciences, Department of Nursing, Chang Gung University of Science and Technology, Chiayi 61363, Taiwan; ccteng@mail.cgust.edu.tw; 6Research Fellow, Chang Gung Memorial Hospital, Chiayi 61363, Taiwan; 7Department of Pathology, Chang Gung Memorial Hospital, Chiayi 61363, Taiwan; lkf2002@cgmh.org.tw; 8Research Center for Food and Cosmetic Safety, College of Human Ecology, Chang Gung University of Science and Technology, Taoyuan 33303, Taiwan; 9Chronic Diseases and Health Promotion Research Center, Chang Gung University of Science and Technology, Chiayi 61363, Taiwan

**Keywords:** colorectal cancer (CRC), lymph node metastasis, gelsolin (GSN), peroxiredoxin 4 (PRDX4), EGFR/RhoA/PKCα/ERK

## Abstract

**Simple Summary:**

Schematic presentation of the histone modification of gelsolin (GSN) and peroxiredoxin-4 (PRDX4) promoters by endogenous EGFR/RhoA/PKC/ERK signaling pathways in DLD-1 cells. Identifying lymph node metastasis (LNM)-associated proteins by two-dimensional difference gel electrophoresis MS/MS in colorectal cells (CRC) indicates the reliability of candidate markers of GSN and PRDX4 and therapeutic targets for CRC.

**Abstract:**

Background/Aims: A combination of fluorescence two-dimensional difference gel electrophoresis (2D-DIGE) and matrix-assisted laser desorption/ionization time of flight mass spectrometry approach was used to search for potential markers for prognosis and intervention of colorectal cancer (CRC) at different stages of lymph node metastasis (LMN). This quantitative proteomic survey aimed to investigate the LNM-associated proteins and evaluate the clinicopathological characteristics of these target proteins in CRC from stage I to stage IV. Methods: Sixteen CRC cases were categorized into paired non-LNM and LNM groups, and two-dimensional difference gel electrophoresis and MS proteome analysis were performed. Differential protein expression between non-LNM and LNM CRC was further validated in a tissue microarray, including 40 paraffin-embedded samples by immunohistochemistry staining. Moreover, a Boyden chamber assay, flow cytometry, and shRNA were used to examine the epithelial–mesenchymal transition and mechanism invasiveness of the differentially expressed proteins in DLD-1 cells and in vivo xenograft mouse model. Results: Eighteen differentially expressed proteins were found between non-LNM and LNM CRC tissues. Among them, protein levels of Gelsolin (GSN) and peroxiredoxin 4 (PRDX4) were abundant in node-positive CRC. Downregulation of GSN and PRDX4 markedly suppressed migration and invasiveness and also induced cell cycle G1/S arrest in DLD-1. Mechanistically, the EGFR/RhoA/PKCα/ERK pathways are critical for transcriptional activation of histone modification of H3 lysine 4 trimethylation (H3K4me3) of GSN and PRDX4 promoters, resulting in upregulation of GSN, PRDX4, Twist-1/2, cyclinD1, proliferating cell-nuclear antigen, β-catenin, N-cadherin, and matrix metalloprotein-9. Conclusions: GSN and PRDX4 are novel regulators in CRC lymph node metastasis to potentially provide new insights into the mechanism of CRC progression and serve as a biomarker for CRC diagnosis at the metastatic stage.

## 1. Introduction

Colorectal cancer (CRC), one of the most common cancers worldwide, is frequently categorized as a leading cause of cancer-related deaths [1] due to its uncontrolled metastasis [2]. Although new surgical, radiotherapy techniques and chemotherapy have significantly improved CRC treatment over the recent decade, its overall survival rate has not changed remarkably [3]. Lymph node metastasis is a major factor that contributes to this poor outcome in CRC treatment. Therefore, advanced early diagnosis of CRC is important for preventing the occurrence of metastasis [4]. Unfortunately, there are still no effective and accurate methods for the early diagnosis of metastatic CRC [5]. The presence of positive lymph nodes is a key event that separates stage I from stage IV CRC in patient management; hence, the search for specific protein biomarkers associated with lymph node metastasis could benefit the development of the early detection of CRC metastasis. Although lymph node involvement and metastasis are the key factors in causing the death of patients with CRC, the mechanisms for the development of lymph node metastasis (LNM) in CRC are not fully elucidated [6]. In fact, a series of alterations in the protein expression at multiple steps are involved and required to develop LNM [7]. For instance, the proteins involved in cellular epithelial–mesenchymal transition (EMT) are associated with migratory and invasive traits of cancer cells and has caused an inferior patient survival rate, including N-cadherin [7,8] and downstream of the EGFR/RhoA/PKCα/ERK pathway for the regulation of adherens junctions. In addition, aberrant Wnt/β-catenin activation triggers tumorigenesis through several transcription factors, such as the Twist 1/2 family [9,10,11,12,13]. The upregulation of the levels of transcription factors, including vimentin and MMP-9, suppresses E-cadherin activity, and these promote EMT CRC malignancy [13]. On the other hand, they also promote cell cycle progression by controlling several key proteins, including CDK4 with Cyclin D1 [13,14,15]. However, how EMT induction develops CRC metastasis is still not clear. Meanwhile, the identification of reliable candidate markers of LNM-associated proteins is an urgent issue for the early prognosis of CRC.

Compared with traditional 2D polyacrylamide gel electrophoresis, fluorescence two-dimensional difference gel electrophoresis (2D-DIGE) [16] offers higher sensitivity, accuracy, and resolution as an advanced quantitative proteomics technology. Prior to separation by 2-DE, this method is used to pre-label different protein samples with fluorescent cyanine dyes (Cy2, Cy3, and Cy5). Therefore, different dyes were used to label different samples, which were separated in the same 2D gel. Meanwhile, to avoid inter-gel variation, the same internal standard can also be used in every gel to obtain an accurate and reproducible quantitation of differences between stage I and stage IV CRC tissues by 2D-DIGE. In order to examine the expression profiles of the LNM CRC-associated proteins involved in cell migration and invasiveness, proteomic analysis was performed using 2D-DIGE LC-MS/MS with tissue microarray [17]. In this study, Gelsolin (GSN) and peroxiredoxin 4 (PRDX4) proteins associated with LNM in CRC were identified by 2D-DIGE MS. We investigated whether GSN and PRDX4 influenced invasion, survival, and EMT in DLD-1 cell lines. Furthermore, GSN and PRDX4 levels were upregulated by the intracellular signaling cascades of the EGFR/RhoA/PKCα/ERK pathways, such as H3 lysine 4 (H3K4) methylation (H3K4me3)-mediated transcription activation of GSN and PRDX4. Moreover, GSN and PRDX4 expressions are essential for the proliferation and metastasis of the DLD-1 cells by the EGFR/RhoA/PKCα/ERK signaling pathways. When taken together, the identification of GSN and PRDX4 proteins associated with LNM in CRC may not only be used as novel biomarkers for the early prognosis of CRC metastasis but also act as promising targets for the treatment of cancer growth, migration, and invasion.

## 2. Materials and Methods

### 2.1. Tissue Samples and Assessment of CRC Tissue Microarrays (TMAs)

CRC specimens paired with corresponding normal tissue were obtained from Chang Gung Memorial Hospital, Chiayi, Cancer Tissue Bank, Taiwan. Sixteen patients with CRC (eight stage I node-negative, eight stage IV node-positive), who underwent surgery but had no presurgical chemotherapy or radiation therapy from 2003 to 2006, were included in this study (IRB103-2992C/CGMH; Table 1). Tissues were snap-frozen in liquid nitrogen and stored at −80 °C. Two independent pathologists confirmed the diagnosis of all samples used in the study. TMAs used in the confirmatory studies represented 40 cases of eligible CRC specimens (18 node-negative, 22 node-positive) and were surgically resected at Chang Gung Memorial Hospital, Chiayi, Taiwan (Appendix A). The excised samples were obtained and previously described [18]. The TMA used in the confirmatory studies represented 40 cases of eligible CRC specimens and was used for immunohistochemistry with anti-GSN and PRDX4 antibodies. The Ethics Committee of Chang Gung Memorial Hospital and our institute approved the protocol. Within one hour of surgery, the excised polyp and cancer samples were collected for further analyses. Immunostaining was assessed independently by a pathologist in a blinded manner. The staining of Peroxiredoxin 4 and Gelsolin were scored as the product of the staining intensity (on a scale of 0–2: negative = 0, low = 1, high = 2) and the percentage of cells stained (on a scale of 0–3: 0 = zero, 1 = 1–25%, 2 = 26–50%, 3 = 51–100%), resulting in scores on a scale of 0–6. After the pathologist confirmed the diagnosis of all samples, the TMA was used in duplicate [18] for immunohistochemical analysis and then independently assessed and scored according to the protocol in our previous publication [19].

### 2.2. Chemical Reagents and Antibodies

All culture materials were purchased from Gibco (Grand Island, NY, USA). 3-(4,5-dimethylthiazol-2-yl)-2,5-diphenyltetrazolium bromide (MTT), protein kinase Cα inhibitor (G06976), epidermal growth factor receptor (EGFR) inhibitor (CAS879127-07-8), extracellular signal-regulated kinase ½ (ERK1/2) inhibitor (PD98059), and RhoA GTPase inhibitor (CCG-1423) were purchased from Sigma (St. Louis, MO, USA). Rabbit polyclonal antibodies (diluted 1:1000) against GSN and PRDX4 were purchased from GeneTex Inc. (GeneTex, TX, USA). Mouse monoclonal antibodies (diluted 1:1000) against Matrix metalloproteinase-9 (MMP-9), β-catenin, N-cadherin, proliferating cell nuclear antigen (PCNA), H3K4me3, and β-actin were purchased from Santa Cruz Biotechnology (Santa Cruz, CA, USA). Rabbit polyclonal antibodies (diluted 1:1000) against RhoA Ser^188^ were purchased from Abcam Technology (Abcam, Boston, MA, USA). Rabbit monoclonal antibodies against EGF receptor Tyr^1^^068^, p44/42 mitogen-activated protein kinase (MAPK) (Erk1/2) Thr^202^/Tyr^204^, and PKCα/β II Thr^638/641^ monoclonal antibodies were purchased from Cell Signaling Technology (Beverly, MA, USA).

### 2.3. Protein Extraction, Protein Labeling, and Gel Image Analysis

Tissue pieces were put into ceramic beads-containing special centrifuge tubes (Roche, Penzberg, Germany); then, 500 μL of pre-chilled Lysis buffer (8 M urea, 4% CHAPS, 30 mM Tris-Cl, pH 8.5) was immediately added and subjected to oscillation by the MagNA Lyser machine at 6500 r/min, thrice. The lysate was then centrifuged at 10,000× *g* for 30 min at 4 °C, and the supernatant was saved and assayed for protein concentration. Total protein was quantified in 10 pooled control samples and 10 pooled patient samples using the Bio-Rad protein assay. The proteins were then concentrated, and impurities were removed using the 2D Clean Up Kit. The final precipitated pellets were redissolved in a lysis buffer (8 M urea, 4% CHAPS, 30 mM Tris-Cl, pH 8.5), and the total protein was quantified again. The samples were stored on ice prior to subsequent Cy dye labeling. Each sample was labeled with 200 pmol (1 μL) of GE CyDye DIGE Fluor Labeling Kit (GE Healthcare, Amersham, UK), incubated on ice for 30 min in the dark and centrifuged at 12,000 rpm for 10 min at 4 °C. This was followed by quenching of the CyDye labeling activities by adding 1 μL of 10 mmol/L lysine to 50 μg of proteins for 10 min on ice in the dark, according to the manufacturer’s protocol. The DIGE gels were scanned using a Typhoon TRIO laser scanner (GE Health Care Life Sciences). The excitation and emission were estimated at 532/580 nm (Cy3), 633/670 nm (Cy5), and 488/520 nm (Cy2). The data of the protein spots were analyzed using DeCyder software (*p*  <  0.05 in the t-test; the ratio of patient/normal >1.5 or <1.5) in 10 patients and 10 control samples [20].

### 2.4. Protein Identification by Mass Spectrometry (MS)

The gels were silver stained, and the protein spots were excised from the gel and washed thrice with H_2_O for 5 min. After washing, the gels were destained twice in a solution [50 mM NH_4_HCO_3_/100% ACN (3:2)] for 15 min each. The supernatant was discarded, and 200 μL of 100% ACN was added by vortexing for 10 min. Proteins were then reduced in 10 mM DTT in 100 mM ammonium bicarbonate for 30 min at 60 °C. The reduced proteins were then alkylated in 55 mM IAA in 100 mM ammonium bicarbonate for 20 min in the dark. Gels were washed with 50% acetonitrile in 100 mM ammonium bicarbonate three times, 20 min each. Gels were dried in a speed vacuum for 15 min after the dehydration step with 100% acetonitrile for 15 min. After adding 6–8 μL of trypsin (25 ng/μL trypsin, 25 mM NH_4_HCO) to cover the gels, the gels were digested in trypsin solution for 16 h at 37 °C. Peptides were extracted with 50% and 100% acetonitrile. MS spectra and MS/MS fragment ion mass were determined with a Bruker MALDI-TOF/TOF analyzer (Bruker Daltonics, Bremen, Germany), and all product ions were submitted to a computer database and analyzed with the MASCOT MS/MS ion search (Matrix Science Inc., MA, USA) using the SwissProt database (all entries). The Bruker operation system, namely flexControl Version 3.3 (Build 85) and flexAnalysis software, were used to create peak lists [21].

### 2.5. Human Colon Cancer (CRC) Cell Cultures and Maintenance

CRC cell line DLD-1 (CCL-221) was obtained from American Type Culture Collection (ATCC) and cultured in RPMI 1640 medium and DMEM medium, respectively, at 37 °C with a humidified 5% CO2 incubator as described in a previous publication [22].

### 2.6. Generation of Stably and Transiently Expressing Gelsolin and Peroxiredoxin-4 Colorectal Cancer Cell Clone

After transfection with GSN and PRDX4 short hairpin RNA (shRNA) (sc-37330-V and sc-40835-V) or scrambled shRNA (sc-108084; purchased from Santa Cruz Biotechnology) lentiviral particles in 0.5 mL of serum-free media, DLD-1 cells were selected for picking up colonies, and the target protein level was assessed as previously described [23].

### 2.7. Cell Cycle Distribution Analysis

In this study, 4′,6-diamidino-2-phenylindole (DAPI)-stained cells were used to examine the changes in cell morphology during apoptosis under fluorescence microscopy. The cells were fixed with 4% paraformaldehyde for 30 min at room temperature and permeabilized with 0.2% Triton X-100 in phosphate-buffered saline, then incubated with 30 min DAPI (1 μg/mL). In order to check the percentage of apoptotic nuclei, 200 to 300 cells were scored using a fluorescent microscope. Cell-cycle distribution of the cells was analyzed by propidium iodide staining and FACS analysis (Attune NxT Flow Cytometer, Thermo Fisher Scientific Inc., Waltham, MA, USA).

The fluorescent probes of 10 μM 2′, 7′-dichlorofluorescein diacetate (H2DCFDA, Molecular Probes) were used to measure hydrogen peroxide (H_2_O_2_), which represented the intracellular accumulation of ROS. The cells with H2DCFDA staining were subjected to a FACS analysis (Attune NxT Flow Cytometer, Thermo Fisher Scientific Inc., Waltham, MA USA). The data of the fluorescent intensity and the number of stained cells were quantified and analyzed and were represented as a percentage of the untreated control group in three independent experiments [24].

### 2.8. Preparation of Total Cell Extracts and Immunoblot Analyses

The cells were lysed with a buffer comprising 1% NP-40, 0.5% sodium deoxycholate, 0.1% sodium dodecyl sulfate (SDS), and a protease inhibitor mixture (phenylmethylsulfonyl fluoride, aprotinin, and sodium orthovanadate); the protein lysates were obtained, as previously described. After electrophoresis, the proteins in SDS-polyacrylamide gel (12% running, 4% stacking) were transferred to a PVDF membrane, and their expression was detected using specific antibodies in the Western light chemiluminescent detection system (Bio-Rad, Hercules, CA, USA). Using the ImageGauge 3.46 software (Fujifilm, Inc., Kanagawa, Japan) as previously described, the area of the photo images in immunoblot for quantitative analysis was measured in terms of their numbers of pixels [25].

### 2.9. Matrigel Invasion and Scratch Assays

As described previously [26], a straight wound line was made in the monolayer of the transfected cells, and the pictures of the wound line were analyzed using the Openlab v3.0.2 image analysis software (Improvision, Coventry, UK). The Boyden chamber was used to detect Matrigel invasion of tumor cells as previously described [26,27]. After 24 h of incubation, the membrane of the cells was fixed and stained with the modified Giemsa stain (Sigma-Aldrich, Burlington, MA, USA). Then, the average number/field of the cells on the lower side of the membrane from triplicated wells with five random fields was analyzed and plotted.

### 2.10. Establishment of a Subcutaneous Tumor Xenograft

Animal care and the general protocols for animal experiments were approved by the Institutional Animal Care and Use Committee of Chang Gung Memorial Hospital, Chiayi, Animal Ethics Research Board (IACUC approval: 2019032502). Male BALB/c-nu nude mice, 4–6 weeks old (18–20 g), were purchased from the National Laboratory Animal Center in Taiwan and maintained under specific pathogen-free (SPF) conditions with sterilized food and water. The DLD-1 cells (10^6^ cells/0.2 mL) were injected subcutaneously into the flanks of 4-week-old to 6-week-old female athymic BALB/c-nu mice. After tumor inoculation, the mice were randomly categorized into four groups (n = 6 per group). The control group animals were treated daily with 0.1 mL PBS (i.p.). Tumor volumes were monitored and measured using calipers. After 18 days of treatment, the mice were euthanized, and their tumors and organs, including the liver, lungs, and kidneys, were collected for further analysis. Tumor tissue sections were fixed in 4% formaldehyde and then embedded in paraffin blocks. Subsequently, further histochemical and immunohistochemical analyses were conducted, as described previously.

### 2.11. Chromatin Immunoprecipitation (ChIP) Analysis

DLD-1 cells were incubated with 1% formaldehyde at room temperature to generate a DNA-protein cross-link. After 10 min, 125 mM glycine was added to the cells for 5 min. The cells were scraped and placed in a sodium dodecyl sulfate (SDS) lysis buffer (50 mM Tris-HCl [pH 8.1], 1% SDS, and 10 mM EDTA) and rotated with specific antibodies against H3K4me3 and 2 μL of non-immunized rabbit IgG, as ‘no antibody’ negative control, overnight at 4 °C in the presence of protease inhibitors (1 μg/mL leupeptin, aprotinin, and pepstatin A, 1 mM phenylmethylsulfonyl fluoride [PMSF]). After elution with an elution buffer (50 mM Tris-Cl [pH 7.5], 1 mM EDTA, 1% SDS), the cross-linking of immunoprecipitated complexes was reversed by incubating at 65 °C for at least 2 h. DNA fragments were purified using a ChIP DNA Clean & Concentrator Kit (Zymo), and quantitative PCR was then performed to amplify the promoter region of the GSN and PRDX4 genes using specific primers (GSN-607 to -407 bp: forward, 5′-TAGTCGATGTCTGGGGTTCC-3′ and reverse 5′-AAAGCCCTCGAAGAGTGACA-3′; and PRDX4: forward, 5′-CAAATGCAGGCTTGGGATGG-3′, and reverse 5′-CAGCGCCTCCATGACCACG-3′) under the following conditions: 40 cycles of denaturation at 94 °C, annealing at 60 °C, and extension at 72 °C. After each PCR, disassociation curves were generated to ensure that a single amplified PCR product of appropriate length was resolved by electrophoresis. The test samples may also be expressed as a percentage of a reference gene: that is, by calculating the percent of input for each ChIP: % input = 2(−ΔCt [normalized ChIP]); normalizing the positive locus ΔCt values to negative locus (ΔΔCt) by subtracting the ΔCt value for the positive locus from the ΔCt value for negative locus: (ΔΔCt = ΔCt positive − ΔCt negative). The fold enrichment of the positive locus sequence in ChIP DNA over the negative locus was calculated as Fold enrichment = 2ΔΔCt. In addition, the mean CT ± SE was calculated from individual CT values and obtained from triplicate determinations at each stage. The normalized mean CT was estimated as ΔCT by subtracting the mean CT of the input from that of the individual region among the untreated control group and the drug treatment groups. Specific genes were calculated by the ΔΔCt method: 2^(−ΔΔCt[treatment/control])^, where ΔΔCt[treatment/control] = (∆Ct[treatment]) − (∆Ct[control]). The results were statistically analyzed using the Student’s paired *t*-test. A *p*-value of <0.05 was considered to be statistically significant [27,28].

### 2.12. Statistical Analysis

The results were expressed as mean ± standard deviation of three repeats from more than triplicate independent experiments. Statistically significant differences were evaluated and established at *p* < 0.05 by one-way ANOVA with a post hoc Mann–Whitney *u* test and Student’s paired *t*-test using the SPSS software (version 10.0; SPSS, Chicago, IL, USA) [18,29].

## 3. Results

### 3.1. Identification of Differentially Expressed Proteins Associated with Lymph Node Metastasis Stage I and Stage IV of Human Colorectal Cancer by Two-Dimensional Difference Gel Electrophoresis Proteome Map

The 2D-DIGE approach in a nonlinear pI range of 3.0–10.0 and a molecular weight range of 10 to 120 kDa was used to analyze the proteomes of the CRC samples from eight node-positive and eight node-negative patients (Table 1). Technically, we performed a DIGE gel with an independent proteome preparation from a paired LMN stage I tissue tumors stained with Cy3 (green), from LMN stage IV tissue tumors stained with Cy5 (red), and from a mixture of all protein samples as the internal standard stained with Cy2 (blue) to understand their differential protein profiling systematically (Figure 1). The black-and-white scanning images of each stained gel are shown in the right panel of Figure 2 and analyzed by using DeCyder software. The expression level of a total of 18 spots was significantly changed (***** *p* < 0.05) among all the CRC samples. Compared with the corresponding node-negative tissues, the CRC groups exhibited a greater abundance of 9 protein spots and a lower abundance of 9 spots. These spots of the differentially expressed proteins are indicated on the gel images (Figure 1).

### 3.2. Proteomic Profiling of the Lymph Node Metastasized Colorectal Cancer Tissues by Mass Spectrometry

Net, we investigated the novel protein associated with CRC LMN stage I and stage IV [27,28] using proteomic 2D-DIGE with the LC-MS/MS analysis. More than 800 protein spots from cell lysates of LMN tissue tumors were visualized by silver-stained GE analysis and analyzed by ImageMaster software (Figure 2). Our analysis showed 18 differentially expressed proteins, with more than a 1.5-fold change in their expression levels, in comparison with those in the LMN stage I and stage IV groups. Among them, nine proteins had a greater expression in the LMN stage I than in stage IV (Table 2). By contrast, the intensity of nine protein spots increased consistently in the LMN stage IV (Table 2). The full names of proteins after the peptide fingerprint identification of 18 spots by MALDI-TOF MS are listed. Bioinformatics analysis by MetaCore™ software differentially expressed proteins were further characterized in order to understand the underlying protein–protein interaction networks and to evaluate candidate proteins as potential biomarkers. Potential interaction and regulation network was associated with these proteins regarding cellular signaling and gene expression, which was distributed in the extracellular space, membrane, and cytoplasm (Figure 3). We found that Gelsolin and PRDX4 are linked together, whose levels are influenced as a prognostic and predictive biomarker in CRC tumor tissues with lymph node metastasis (LNM) stage IV. In the zoomed views of a representative gel region, several differentially expressed, oxidative stress-related proteins are shown, including apolipoprotein A-I (spot 10), GSN (spot 15), PRDX4 (spot 16), and proteasome activator complex subunit 2 (spot 17). Proteins in spots 15 and 16 were also identified by using a 2D-DIGE proteomic analysis. Interestingly, GSN and PRDX4 may participate in the aggression and progression of CRC cells [30]. For instance, the network analysis apparently shows that GSN and PRDX4 are linked together, indicating their potential use as a prognostic and predictive biomarker in oncology [31]. Indeed, GSN functions as an actin-modulating protein and plays a role in tumorigenesis, invasion, and migration. In addition to physiological functions, PRDX4 is also involved in therapeutic resistance, metastasis, and recurrence of tumors; its high expression is associated with the depth of invasion, lymph node metastasis, and short survival time [32,33]. Noteworthily, this study is the first to determine the roles of GSN and PRDX4 in the migration and invasion of CRC and tumor growth in nude mice xenograft models of cancer.

### 3.3. Validation of the Gelsolin and Peroxiredoxin-4 Expression Patterns on Tissue Microarrays

To confirm the expression levels of GSN and PRDX4 in CRC, we evaluated 40 CRC samples stratified by node status, including 18 node-negative and 22 node-positive patients. Each of the CRC samples was paired with a duplicate at the time of resection. Significance analysis of tissue microarrays (TMAs) in each intergroup was performed. After immunohistochemical staining of TMAs, we further investigated the correlation between the GSN and PRDX4 expressions and node status in a larger, independent patient cohort. Forty eligible cases of the TMAs from two sources with different geographic regions in a limitation were subsequently analyzed. IHC staining data showed that GSN and PRDX4 are mainly expressed in the cytoplasm (Figure 4). Based on a 0–6 scale analysis system representing a combination of staining intensity and fraction of cells stained, our results revealed a high-level GSN and PRDX4 expression in node-positive CRC in this intergroup (Figure 4). A Mann–Whitney U test indicated that the distribution of the scores in the node-negative and node-positive groups was significantly different (Table 3) [19]. These findings demonstrate that high-level expression of PRDX4 and GSN might contribute to CRC progression (*p* = 0.04 and *p* = 0.02; Table 3) and may be a potential target for the treatment of this disease.

### 3.4. Effects of Gelsolin and Peroxiredoxin-4 on Cell Biologic Processes

The functional roles of GSN and PRDX4 in tumor growth, migration, and invasion of CRC remain unclear. Therefore, to determine how GSN and PRDX4 promote metastasis in CRC, short hairpin RNA (shRNA) for knockdown of GSN and PRDX4 or scrambled shRNA lentiviral particles (control group) were transfected into DLD-1 cells, and as a result, the levels of GSN and PRDX4 proteins were decreased, but not by the transfection of scrambled shRNA (Figure 5A). Subsequently, the effects of GSN and PRDX4 on biologic processes relevant to metastasis were determined in the human DLD-1, stably expressing GSN or PRDX4 shRNA (Figure 5A). The eukaryotic cell cycle control system incorporates a complex process and checkpoints that monitor and dictate the progression of cancer [32]. Cell cycle analysis indeed revealed that GSN and PRDX4 shRNAs-transfected cells had arrested at the G1/S stage of the cell cycle (Figure 5B). Inactivation of GSN and PRDX4 induced G1 arrest by 62% and 73%, respectively (Figure 4B). Furthermore, the scratch-wound assay with a lasting rapid movement revealed that the of PRDX4 and GSN by specific shRNAs marly decreased cell migration front resulted in a marked reduction at 48 h in the DLD-1 cells compared with the cells with control shRNA-transfected group (Control shRNA) (^#^^&^
*p* < 0.05, Figure 6A). A Boyden chamber assay to evaluate the in vitro migration and invasion effect of GSN and PRDX4 showed that shRNAs-mediated knockdown of GSN and PRDX4 reduced 40% and 38% of the invasion, respectively, then the cells transfected with the control shRNA (***** *p* < 0.05, Figure 6B). These data suggest that GSN and PRDX4 play pivotal roles in cell proliferation and metastasis of CRC.

### 3.5. Effects of Gelsolin and Peroxiredoxin-4 on Epithelial–Mesenchymal Transition in Colorectal Cancer

EMT is a key developmental process of cancer invasion and metastasis to connect adherens junctions and focal adhesions [7,8]. N-cadherin-mediated adherens junctions as the EMT hallmark facilitates the activation of epidermal growth factor receptor/RhoAGTP interaction, and PKCα/ERK pathways are associated with cancer cell motility. The effects of GSN and PRDX4 on biological processes relevant to the EGFR/RhoA/PKCα/ERK signaling pathways in CRC and the protein levels p-EGFR, p-RhoA, p-PKCα, and p-ERK1/2 in DLD-1 cells were widely used as a model for studies of cell signaling and invasion; therefore their expressions were determined in this study by western blot analysis. GSN and PRDX4 shRNA significantly decreased the expression of the EGFR/RhoA/PKCα/ERK pathway in human DLD-1 (Figure 7). Our results demonstrated that the GSN and PRDX4 shRNA significantly decreased the expression of these EMT-related proteins, such as twist 1/2 and cyclin D1, but their expressions were not affected in the Control shRNA-transfected cells (siControl) or the untreated group (Control) (Figure 6).

### 3.6. Suppression of Colorectal Cancer Tumorigenesis by Loss of Gelsolin and Peroxiredoxin-4 In Vivo

So as to determine whether the knockdown of GSN and PRDX4 can suppress tumorigenesis in vivo, a nude mouse model of CRC cells xenograft (n = 6) was implanted subcutaneously into the DLD-1 cells transfected with the control shRNA (Control shRNA) or the GSN shRNA and PRDX4 shRNA (Figure 8). Tumor tissues were isolated and weighed 18 days after the xenograft implantation. The transfection of the DLD-1 cells with GSN shRNA and PRDX4 shRNA led to significantly decreased tumor weight in tumor tissues compared with that of the DLD-1 cells (Control shRNA) in mouse xenografts (Figure 8A). The protein level of PCNA (a marker for cell proliferation), N-Cadherin (a marker for EMT transition in tumorigenesis), β-catenin (a marker in advanced colorectal carcinoma), and MMP-9 (a key regulator for EMT-related metastatic capability) were further detected in these tumor tissues by immunohistochemistry. Consistent with the in vitro data, their expressions were significantly decreased in the tumors from the DLD-1 cells treated with PRDX4 shRNA and GSN shRNA than that in the Control shRNA group (Figure 8B). Hence, the knockdown of GSN and PRDX4 inhibits the growth of the CRC cells xenograft in vivo.

### 3.7. Histone Modification of Gelsolin and Peroxiredoxin-4 Promoters by EGFR/RhoA/PKC/ERK Signaling Pathways

Pleiotropic signals of EGFR/RhoA/PKCα/ERK activation are involved in EMT activation [9,10]. Therefore, we examined whether this signal can regulate EMT through the regulation of the GSN and PRDX4 expressions in DLD-1 cells. A previous study reported that cellular EGFR/RhoA/PKCα/ERK signaling pathways also influence methylation at H3K4me3 to regulate cell proliferation, growth, and survival [12]. Therefore, ChIP analysis was performed to investigate the role of EGFR/RhoA/PKCα/ERK signaling pathways in histone modification of the GSN and PRDX4 promoter; for this, H3K4me3 was assessed in DLD-1 cells with the following treatments [34]. Our results showed that incubated DLD-1 cells with inhibitors, including protein kinase Cα inhibitor G06976 (5 nM), RhoAGTPase inhibitor CCG-1423 (1.0 μM), ERK1/2 inhibitor PD98059 (50 μM), and EGFR inhibitor CAS879127-07-8 (10 μM), could decrease the trimethylation of H3K4me3 on promoter regions of both PRDX4 and GSN genes. The histone H3 lysine 4 modifications are a well-known marker for transcriptional activation. Altogether, our results show that a synergistic function in H3K4 methylation and transcription activation is required for the upregulation of GSN and PRDX4 in DLD-1 cells by the EGFR/RhoA/PKCα/ERK signaling pathways in CRC survival and metastasis.

## 4. Discussion

CRC, one of the most aggressive cancers, is highly prevalent in most countries [35]. In order to remove this fatal cancer from patients, surgical operations and subsequent chemotherapy and radiotherapy are performed but have a lower overall survival rate. Therefore, it is important to seek novel CRC tumor markers to estimate the malignancy. Detection of positive lymph nodes is a key management step to separate the CRC status from stage I and stage IV CRC tissues for patients. Furthermore, the status of lymph nodes can be ascertained with the number of examined nodes [36]. The heterogeneity of advanced CRC is a major challenge in oncology for their management and treatment. A better understanding of CRC metastasis will lead to the identification of new targets for both early diagnosis and treatment. The current study performed a proteomic analysis of samples of LMN in eight pairs of CRC cancer samples at stage I and stage IV with the goal of identifying new biomarkers by using 2D-DIGE and MS. DIGE-based proteomics with fluorescence labeling exhibits several advantages, including higher sensitivity, reproducibility over a linear dynamic range for better quantification, and fewer technical variations; the process refers to a pooled control as an internal standard (Table 1, Figure 1). In addition, further IHC assays in TMA confirmed the upregulation of GSN and PRDX4 in CRC tissues with LNM stage I and stage IV CRC (Figure 2, Table 2). Eighteen higher differentially expressed proteins were found in all samples; then, GSN and PRDX4 were selected for further study. Indeed, high expression of PRDX4 and GSN might contribute to CRC and correlate with metastatic risk (*p* = 0.04 and *p* = 0.02; Figure 4, Table 3), as indicated by IHC assays in TMA. Functional studies in vitro and silencing of GSN and PRDX4 by lentiviral shRNA showed markedly induced CRC cell cycle arrest in DLD-1 cells (Figure 5). Through a series of experiments, including a Boyden chamber assay, flow cytometry, and western blot assays, we found that GSN or PRDX4 tends to affect the migration and invasion of DLD-1 cells (Figure 6). The 2D-DIGE and MS methods for large-scale protein quantification provide credible means to find potential biomarkers for CRC and its treatment and recurrence. In LNM stage IV of CRC, deregulated GSN and PRDX4 proteins are involved in cancer cell growth and movement, especially invasion and migration. Growth invasion and migration of cancer cells by GSN and PRDX4 play a more important role in recurrent CRC LNM stage IV than in general stage I, suggesting that GSN and PRDX4 promote cell growth, cell cycle distribution, and motility for the progression of human cancers [37]. Increased oxidative stress also drives tumor cell apoptosis for the development of anti-cancer effects [38]. These unique features of GSN and PRDX4 upregulation in tumor cells may play an important role in the oxidative stress response—an anti-apoptotic function. Accordingly, the loss of GSN and PRDX4 increases susceptibility to oxidative stress and tumor aggression and requires further study.

GSN, an actin-binding protein, is a novel target in CRC diagnoses and therapies. The controversial role of GSN has been observed in cancer as both a tumor suppressor gene and oncogene. In the current study, a major role of GSN in the multiple cellular functions like cell motility, apoptosis, morphogenesis, and actin cytoskeletal remodeling was identified. Recently, the GSN level was also found elevated in colon cancer and associated with tumor invasion signaling pathways such as MMP-2/MMP-9 and the MAPK and TGF-β pathways [39,40].

Through the regulation of redox balance, oxidative protein folding, and regulation of hydrogen peroxide signaling, PRDX4, a 2-Cys antioxidant of peroxiredoxins, participates in the progression of breast cancer, prostate cancer, ovarian cancer, colorectal cancer, and lung cancer [41]. Additionally, higher expression of PRDX4 in CRC tissues often correlates with increased liver metastasis [42]. However, how PRDX4 is regulated in CRC remains unclear. The EGFR/RhoA/PKCα/ERK signaling pathway has been previously reported to modulate cancer progression and metastasis through EMT [43,44,45]. In the current study, we established the cell lines with GSN and PRDX4 knockdown and evaluated their biological role in cell invasion, survival, and EMT-associated markers. Interestingly, inactivation of GSN and PRDX4 in cells was significantly associated with a decrease in migratory and invasive potential compared to parental cells in vitro (Figure 5 and Figure 6). Expression levels of GSN and PRDX4 were altered by the inactivation of the EGFR/RhoA/PKCα/ERK pathways and the downregulation of the expression of β-catenin, cyclin D1, twist1/2, PCNA, N-cadherin, and MMP-9 in DLD-1 cells in vitro and in mice in vivo (Figure 7 and Figure 8). Furthermore, experimental manipulation of GSN and PRDX4 expression in established CRC cell lines influenced their invasion, survival, and EMT.

Numerous histone post-translational modifications, such as H3K4me3, are well-established markers of active transcription [34,46,47] that contribute to the oncogenesis and cancer-related functions in the pathogenesis of CRC. In fact, the aberrant methylation of H3K4 not only regulates repair, replication, and transcription-complex proteins but also influences the EGFR/RhoA/PKC PKCα/ERK and Wnt/β-catenin pathways, resulting in proliferation, migration, adhesion, and cell death. Likewise, our data demonstrate that treatment with several inhibitors changed the transcriptional activation of H3K4me3 of the GSN and PRDX4 promoters. Moreover, EGFR/RhoA/PKCα/ERK signaling pathways were observed to play a critical role in GAN and PRDX4-mediated CRC development via methylation at the H3K4 of promoters’ transcriptional activation site in DLD-1 cells (Figure 9). On the other hand, the Wnt/β-catenin signaling pathway also provides a mechanism for the action of GSN and PRDX4, which might broadly regulate the transcription of other genes and influence the processes relevant to the tumor’s microenvironment and metastasis. Further studies are needed to determine the regulatory mechanism for the GSN and PRDX4 signaling axis in the growth, migration, and invasion of CRC cells. Our data demonstrate that GSN and PRDX4 are novel regulators of cellular survival and aggressive pathways in CRC. Furthermore, the EGFR/RhoA/PKCα/ERK signaling pathway participated in the GSN and PRDX4-mediated CRC metastasis via H3K4 methylation.

## 5. Conclusions

Our results demonstrate GSN- and PRDX4-mediated cell survival and aggressive pathways in CRC via the EGFR/RhoA/PKCα/ERK signaling pathways. The expression of GSN and PRDX4 was regulated in the H3K4 site of promoters in DLD-1 cells (Figure 10). Ultimately, this study provides evidence on the molecular mechanism by which GSN and PRDX4 mediate an invasiveness pathway in colorectal cancer cells associated with the pathological stage IV lymph node-positive metastasis of CRC.

## Figures and Tables

**Figure 1 cancers-14-03189-f001:**
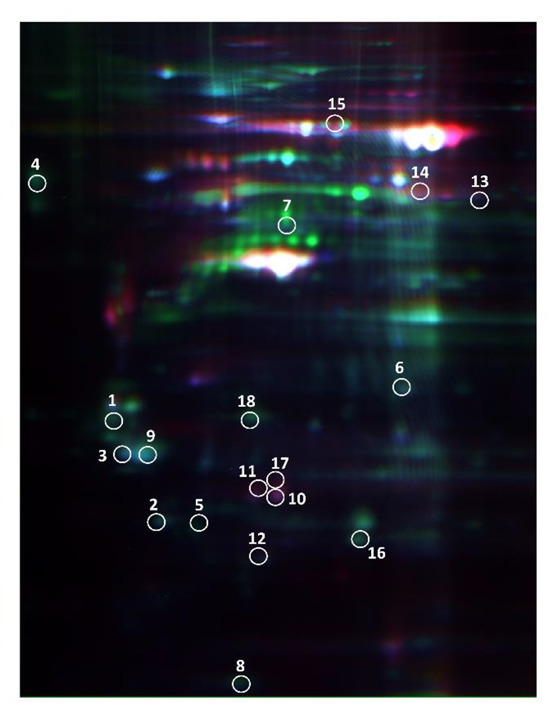
Representative overlapping two-dimensional difference gel electrophoresis (2D-DIGE) expression maps of colorectal cancers stratified by node status labeled with fluorescent dyes (Cy2, Cy3, and Cy5). The 2D-DIGE overlay image of protein spots compares a sample of node-negative patients labeled with Cy3 (green) to a patient sample node-positive labeled with Cy5 (red) and an internal standard sample common to all gels labeled with Cy2 (blue). This image is representative of 1 of the 10 gels analyzed. 1. Tropomyosin beta chain; 2. Translationally-controlled; 3. Proteasome subunit alpha type-5; 4. Glucosidase 2 subunit beta; 5. Transcription activator BRG1; 6. E3 ubiquitin-protein ligase HERC2; 7. Tubulin beta-4B chain; 8. N-alpha-acetyltransferase 35, NatC auxiliary subunit; 9. 14-3-3 protein zeta/delta; 10. Apolipoprotein A-I; 11. Rho GDP-dissociation inhibitor 1; 12. ATP synthase subunit d; 13. Elongation factor Tu; 14. Selenium-binding protein 1; 15. Gelsolin; 16. Peroxiredoxin-4; 17. Proteasome activator complex subunit 2; 18. Heterogeneous nuclear ribonucleoprotein F.

**Figure 2 cancers-14-03189-f002:**
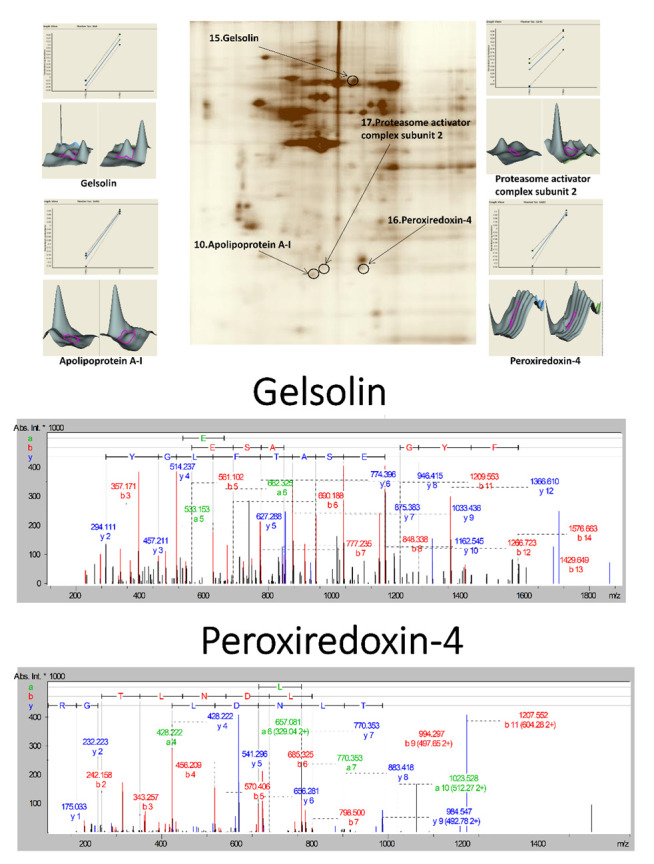
Detailed analyses of gelsolin (GSN) and peroxiredoxin-4 (PRDX4) by two-dimensional difference gel electrophoresis (2D-DIGE). Two-dimensional difference gel electrophoresis gel images for GSN and PRDX4 were labeled as 15 and 16. pI and molecular weight (MW) for spot 15 are 5.8 and 86 kD; pI and MW for spot 16 are 5.8 and 30 kD. Protein waves for spots 15 and 16 in view show both are single peaks, which signifies analysis of the peptide fingerprints. The peptide mass fingerprint of GSN shows the peptides matched to MASCOT search against the SwissProt database. The peptide mass fingerprint of PRDX4 shows the peptides matched to the MASCOT search against the SwissProt database.

**Figure 3 cancers-14-03189-f003:**
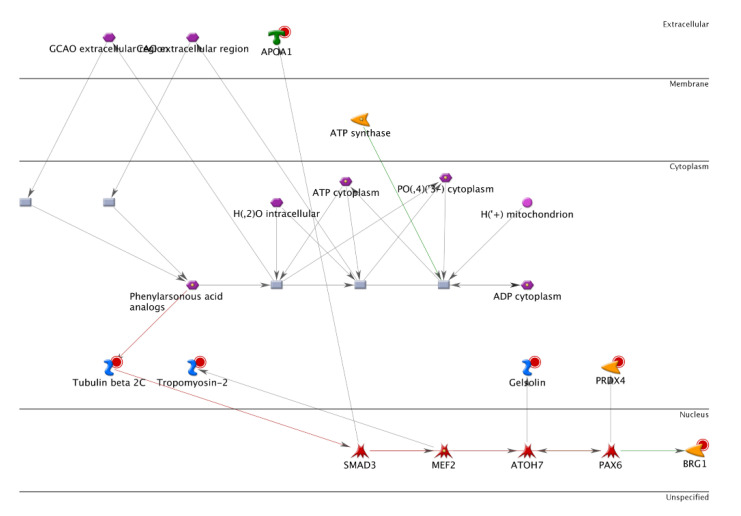
Network of protein interactions associated with eighteen candidate proteins at the cellular and genetic levels by MetaCore. The network was generated using the shortest path algorithm to profile interaction among different proteins. The arrowheads indicated the direction of the interaction. The colorful lines between nodes indicated interactions.

**Figure 4 cancers-14-03189-f004:**
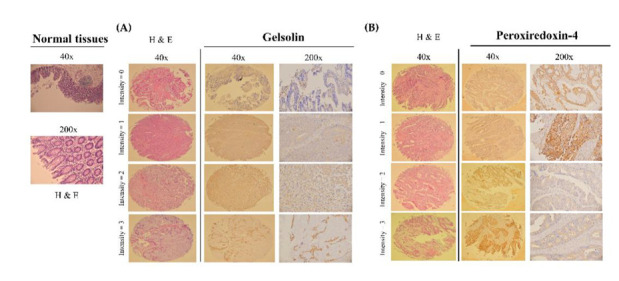
Illustrations of hematoxylin and eosin staining and immunohistochemistry of GSN and PRDX4 on intensity scores of CRC tissue microarrays. (**A**) Gelsolin and (**B**) Peroxiredoxin-4.

**Figure 5 cancers-14-03189-f005:**
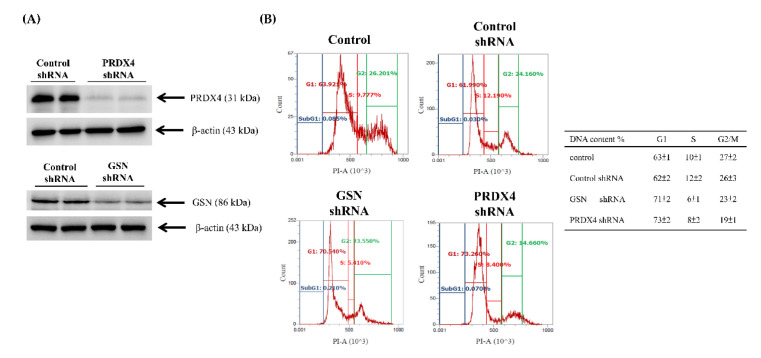
The effect of gelsolin (GSN) and peroxiredoxin-4 (PRDX4) inactivation on cell cycle distribution and oxidative status in DLD-1 cells. (**A**) Lysates from Control, Control shRNA, GSN shRNA, and PRDX4 shRNA clones of DLD-1 cells were used for western blot analysis. Data presented in the western blot are derived from a representative study. (**B**) The GSN shRNA and PRDX4 shRNA cells were fixed and stained with propidium iodide, and the DNA content was analyzed by flow cytometry. The cell number percentage in each phase (G1, S, and G2/M) of the cell cycle was calculated. The original western blots can be found at Appendix A.

**Figure 6 cancers-14-03189-f006:**
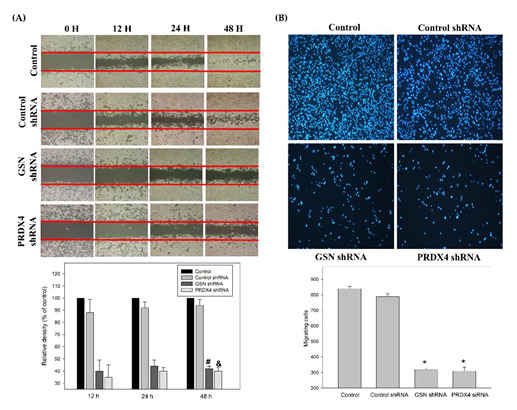
The effects of gelsolin (GSN) and peroxiredoxin-4 (PRDX4) silencing on cell migration and invasiveness of human colorectal cancer cells. (**A**) DLD-1 cells were transfected with lentiviral shRNA targeting GSN and PRDX4 (GSN shRNA and PRDX4 shRNA) or non-targeting control (control shRNA). Meanwhile, the cells were treated for 12, 24, and 48 h using the scratch-wound assay and were visualized. The percentage of the surface area filled by the DLD-1 cells was subsequently quantified by densitometric analyses relative to the control, which was set at 100% in the photograph. Data are presented as means ± SD based on three independent experiments. The experiments were performed in triplicate, and data are presented as means ± SD. ^#^ *p* < 0.05, GSN shRNA compared with the control shRNA group for 48 h (*p* < 0.05) ^&^ *p* < 0.05. PRDX4 shRNA compared with the control shRNA group for 48 h. (**B**) Effect of GSN and PRDX4 knockdown on the invasiveness of DLD-1 cells. Cells were transfected with various shRNA. For 24 h, invasion of the cells through a layer of Matrigel was determined by the Boyden Chamber method, as described in the “Methods” section. The lower and upper chemotaxis cells were separated by a polycarbonate membrane. Microscopy images detected cells that migrated into the inner membrane. Magnification: ×200. The cell invasion was quantified by counting the number of cells that migrated into the inner membrane. The control cells were untreated. The experiments were performed in triplicate, and data are presented as means ± SD. The symbol ***** refers to means that are significantly different when compared to the control shRNA group with *p* < 0.05.

**Figure 7 cancers-14-03189-f007:**
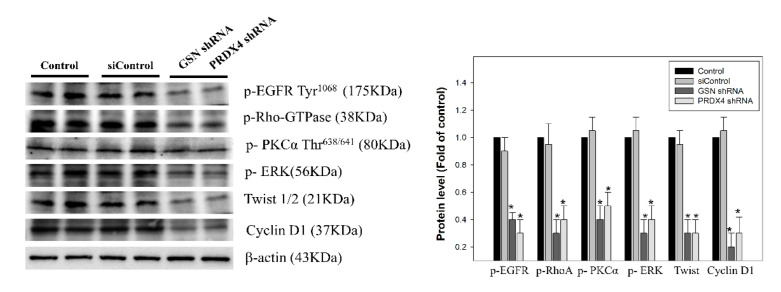
The effect of gelsolin (GSN) and peroxiredoxin-4 (PRDX4) silencing on the activation of the EGFR/RhoA/PKCα/ERK pathway and expression of EMT markers in CRC. DLD-1 cells were transfected with lentiviral shRNA targeting GSN and PRDX4 (GSN shRNA and PRDX4 shRNA) or non-targeting control (Control shRNA). Whole-cell lysate proteins were prepared and analyzed by western blot, with β-actin serving as the loading control. Cell lysates were analyzed by 10% sodium dodecyl sulfate-polyacrylamide gel electrophoresis and subsequently treated with antibodies against phosphorylation of EGFR, RhoA, PKC PKCα, and ERK as well as twist 1/2, and cyclin D1. The protein levels were quantified through densitometric analysis with the ratio of the untreated control (control) set as 1-fold. The quantitative data are presented as the mean of three repeats from three independent experiments. ***** *p* < 0.05, compared with the control shRNA group (siControl).

**Figure 8 cancers-14-03189-f008:**
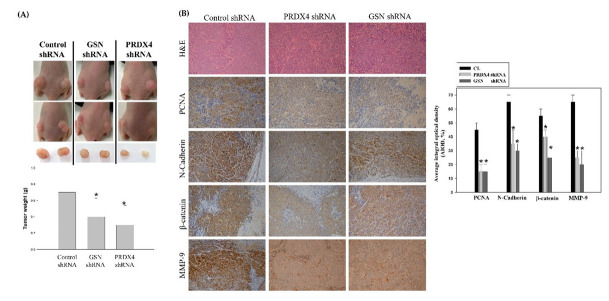
The loss of gelsolin (GSN) and peroxiredoxin-4 (PRDX4) significantly suppressed cell proliferation in the in vivo nude mouse model of colorectal cell (CRC) xenograft. (**A**) Nude mice were injected subcutaneously with 1 × 10^7^ cells/mouse for each indicated Control shRNA, GSN shRNA, and PRDX4 shRNA DLD-1 cell lines. Left panels: the results show the images of nude mice implanted with CRC cancer cells, the tumor volume, and the tumor weight of the quantitative data. Data were expressed as mean ± SD (n = 6/group). ***** *p* < 0.05, compared with the control shRNA group. (**B**) Hematoxylin and eosin stain (1st row, upper panel), the measurement of the protein levels of proliferating cell nuclear antigen (PCNA; 2nd row, upper panel), N-Cadherin (3rd row, upper panel), b-catenin (4th row, upper panel), and matrix metalloprotein-9 (MMP-9; 5th row, upper panel), using an immunohistochemical analysis; the bottom panel: quantitative immunohistochemical proteins, PCNA, N-Cadherin, β-catenin, and MMP-9 stain were evaluated by calculating the average of Integrated Optical Density (AIOD). Multiple tumor fields were evaluated per treatment group. The positively stained area was evaluated from three randomly-selected observational fields of each section. The data were expressed as mean ± SD. (n = 6/group). ***** *p* < 0.05, compared with the control shRNA group, at a magnification of ×200.

**Figure 9 cancers-14-03189-f009:**
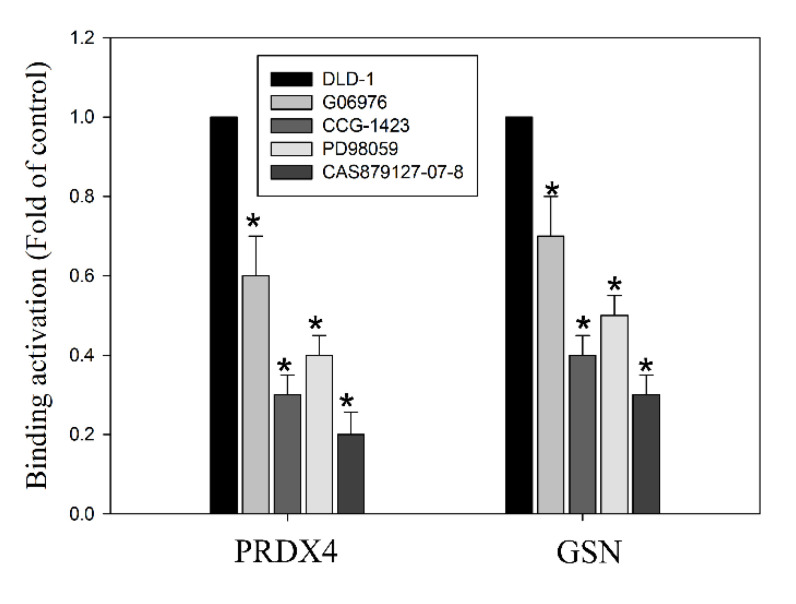
The effect of the kinase inhibitors in blocking histone modification of peroxiredoxin-4 (PRDX4) and gelsolin (GSN) promoters. DLD1 cells were incubated with various concentrations of the specific inhibitors G06976, CCG-1423, PD98059, and CAS879127-07-8 for 24 h. Then, the chromatin immunoprecipitation (ChIP) assays were performed using antibodies against histone H3K9K14ac (Acetyl Lys9/Lys14) to pull down associated DNA. The precipitated DNA was PCR-amplified using primer sets specific to the target sites (−530/−375) of PRDX4 and GGSN promoters. The amount of immunoprecipitated DNA in each sample is represented as a normalized signal to the negative control ChIP. The normalized mean CT was estimated as ΔCT by subtracting the mean CT of the input from that of the individual region among the untreated control group. The effect of the specific inhibitors on specific genes was calculated using the ΔΔCt method. The data are presented as the mean ± SD of three independent experiments. ***** *p* < 0.01, compared to the untreated group.

**Figure 10 cancers-14-03189-f010:**
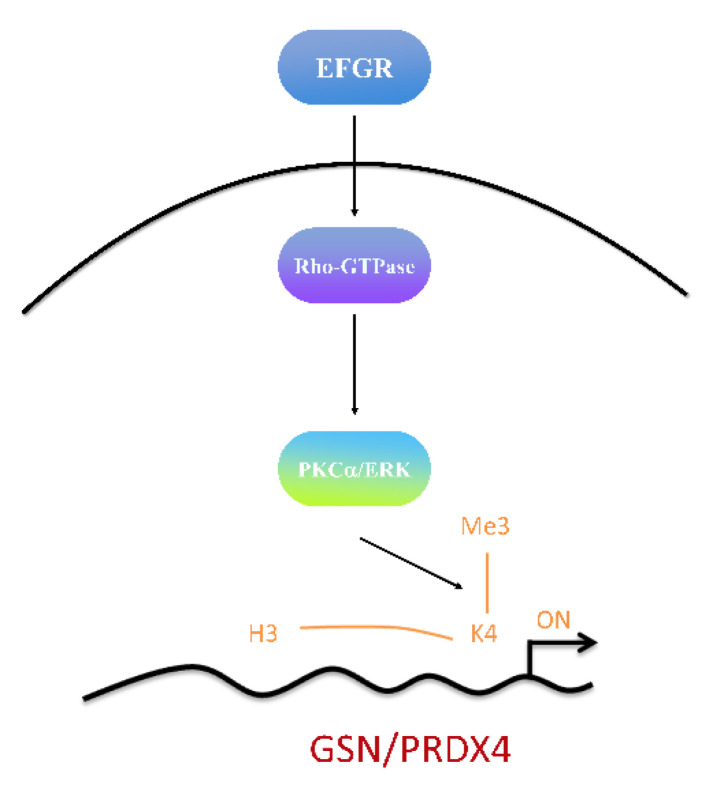
Schematic presentation of the histone modification (H3K4me3) of gelsolin (GSN) and peroxiredoxin-4 (PRDX4) promoters by endogenous EGFR/RhoA/PKC/ERK signaling pathways in DLD-1 cells. Identifying lymph node metastasis (LNM)-associated proteins by two-dimensional difference gel electrophoresis (2D-DIGE)-MS/MS in colorectal cells (CRC) indicates the reliability of candidate markers of GSN and PRDX4 and therapeutic targets for CRC.

**Table 1 cancers-14-03189-t001:** Baseline characteristics of patients with colorectal cancer.

	Stage I (Non-Metastasis; n = 8)	Stage IV (Metastasis; n = 8)
Age (years)	64	64
Sex		
Male	3	5
Female	5	3
Histologic grade		
Well	5	0
Moderate	3	6
Poor	0	2
Primary site		
Proximal colon	1	4
Distal colon	3	2
Rectum	4	2
Primary tumor		
T1	3	0
T2	5	1
T3	-	4
T4	-	3
Lymph node metastasis		
nil	8	0
N1(1–3)	0	2
N2 (>4)	0	6

**Table 2 cancers-14-03189-t002:** Differentially expressed protein for late-stage CRC. Differentially expressed protein (LMN from stage I > stage IV).

Spot	Protein Name	Mr/PI	Accession No	MASCOT Score	Matched Peptides
1	Tropomyosin beta chain	32/4.5	TPM2_HUMAN	304	15
2	Translationally-controlled	19/4.6	TCTP_HUMAN	125	6
3	Proteasome subunit alpha type-5	26/4.5	PSA5_HUMAN	71	71
4	Glucosidase 2 subunit beta	60/4.1	GLU2B_HUMAN	75	3
5	Transcription activator BRG1	85/8.4	SMCA4_HUMAN	42	4
6	E3 ubiquitin-protein ligase HERC2	53/5.8	HERC2_HUMAN	41	4
7	Tubulin beta-4B chain	50/4.6	TBB4B_HUMAN	472	18
8	N-alpha-acetyltransferase 35, NatC auxiliary subunit	84/6.6	NAA35_HUMAN	51	4
9	14-3-3 protein zeta/delta	27/4.5	1433Z_HUMAN	268	14
10	Apolipoprotein A-I	30/5.4	APOA1_HUMAN	350	18
11	Rho GDP-dissociation inhibitor 1	23/4.8	GDIR1_HUMAN	199	9
12	ATP synthase subunit d	18/5.0	ATP5H_HUMAN	72	4
13	Elongation factor Tu	49/7.9	EFTU_HUMAN	323	15
14	Selenium-binding protein 1	52/5.9	SBP1_HUMAN	149	5
15	Gelsolin	86/5.8	GSN_HUMAN	330	17
16	Peroxiredoxin-4	30/5.8	PRDX4_HUMAN	65	4
17	Proteasome activator complex subunit 2	27/5.4	PSME2_HUMAN	186	8
18	Heterogeneous nuclear ribonucleoprotein F	45/5.2	HNRPF_HUMAN	95	4

**Table 3 cancers-14-03189-t003:** Summary of PRDX4 and GSN expression on tissue microarrays.

No.	Protein	Staining (Score)	Node-Negative CRC (n)	Node-Positive CRC (n)
1	Peroxiredoxin-4	None (0)	0 (0.0%)	3 (13.6%)
		Low (1–3)	12 (85.7%)	4 (18.2%)
		High (4–6)	2 (14.3%)	15 (68.2%)
		Total (n)	14 (100%)	22 (100%)
		*p*	0.04	
2	Gelsolin	None (0)	0 (0.0%)	1 (4.5%)
		Low (1–3)	13 (92.9%)	12 (54.5%)
		High (4–6)	1 (7.1%)	9 (40.9%)
		Total (n)	14 (100%)	22 (100%)
		*p*	0.02	

## Data Availability

The data presented in this study are available in this article and Appendix A.

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
