# Peer review of "2D-DIGE-MS Proteomics Approaches for Identification of Gelsolin and Peroxiredoxin 4 with Lymph Node Metastasis in Colorectal Cancer"

_cancers, 2022, doi:10.3390/cancers14133189_

Round 1
Reviewer 1 Report
Reviewer Comments:
Cheng-Yi Huang and coworkers present the manuscript entitled “2D-DIGE-MS Proteomics Approaches for Identification of Gelsolin and Peroxiredoxin 4 with Lymph Nodem Metastasis in Colorectal Cancer”. This is a very complete and interesting study well conducted. However, I have some concerns that must be fully addressed before potential publication of manuscript.
Major comments:
1. Authors should include the graphical abstract and simple summary.
2. Please add the number of the Ethics Committee decision that approved the protocol.
3. In line 136, 500 mL of pre-cooled lysis buffer (8 M urea, 4% 1CHAPS, 30 mM Tris-Cl, pH 8.5) was added, is that correct?
4. The Figure 1. Representative overlapping two-dimensional difference gel electrophoresis (2D-DIGE) ts very nice, However, its difficult to read because the protein names therein. I suggest to enlist the proteins with numbers and then place the names of proteins out of the gel to obtain a clearer image. Also include the MW of detected proteins.
5. Authors should include the dilutions used for each antibody in Western blot analyses.
6. Why did the authors focus on studying only GSN (spot 15) and PRDX4 (spot 16) proteins, knowing their role in tumorigenesis, invasion, and migration in cancer?
7. The authors should perform bioinformatics analysis of the 18 proteins detected in order to elucidate their role in patients with lymph node metastasis in colorectal cancer.
8. In Figure 3. Illustrations of hematoxylin and eosin staining and immunohistochemistry of GSN and 352 PRDX4 on CRC tissue microarrays, please include the normal tissues as controls. Also, the legend is confuse, as authors indicate “Intensity”, but in the table they indicated none, low and high as scores. Also, the Node positive and negative were not indicated. Please correct the figure 3 as in Table 3.
9. The western blots in Figure 4A display very poor quality. Multiple immunodetected bands are showed. I strongly suggest to improve the western blots for both GSN and PRDX4 proteins in order to conclude.
10. Should the authors include immunoblot or qRT-PCR assays to confirm GSN and PRDX4 expression levels in CRC?
11. The authors should include in vivo tumor detection assays by PET imaging or zebrafish xenograft models to evaluate the effect of GSN and PRDX4 on metastasis in CRC.
Author Response
Comments and Suggestions for Authors
Reviewer Comments:
Cheng-Yi Huang and coworkers present the manuscript entitled “2D-DIGE-MS Proteomics Approaches for Identification of Gelsolin and Peroxiredoxin 4 with Lymph Nodem Metastasis in Colorectal Cancer”. This is a very complete and interesting study well conducted. However, I have some concerns that must be fully addressed before potential publication of manuscript.
Major comments:
- Authors should include the graphical abstract and simple summary.
Response: We appreciate this remark of reviewer #1’s comment and respond to the comment, including the graphical abstract and simple summary.
Graphical Abstract
Schematic presentation of the histone modification (H3K4me3) of gelsolin (GSN) and peroxiredoxin-4 (PRDX4) promoters by endogenous EGFR/RhoA/PKCa/ERK signaling pathways in DLD-1 cells. Identifying lymph node metastasis (LNM)-associated proteins by two-dimensional difference gel electrophoresis (2D-DIGE)-MS/MS in colorectal cells (CRC) indicates the reliability of candidate markers of GSN and PRDX4 and therapeutic targets for CRC.
- Please add the number of the Ethics Committee decision that approved the protocol.
Response: We appreciate this remark of comment and these are added in the line 108 Materials and Methods of manuscript. (IRB103-2992C/CGMH; Table 1).
- In line 136, 500 mL of pre-cooled lysis buffer (8 M urea, 4% 1CHAPS, 30 mM Tris-Cl, pH 8.5) was added, is that correct?
Response: We appreciate this remark of comment and these are revised in the line 136 Materials and Methods of manuscript. (500 mL of pre-chilled Lysis buffer).
- The Figure 1. Representative overlapping two-dimensional difference gel electrophoresis (2D-DIGE) ts very nice, However, its difficult to read because the protein names therein. I suggest to enlist the proteins with numbers and then place the names of proteins out of the gel to obtain a clearer image.
Response: Thanks for the comments and these are revised in Figure 1 of manuscript.
- Authors should include the dilutions used for each antibody in Western blot analyses.
Response: We appreciate this remark of comment and these are revised each antibody (diluted 1:1000) in the Materials and Methods of manuscript, line 126/131.
- Why did the authors focus on studying only GSN (spot 15) and PRDX4 (spot 16) proteins, knowing their role in tumorigenesis, invasion, and migration in cancer?
Response: Thanks for the comments. Our analysis using proteomic 2D-DIGE with the LC-MS/MS analysis showed 18 differentially expressed proteins, with more than a 1.5-fold change in their expression levels, in comparison with those in the LMN stage I and stage IV groups. And then the role of the 18 identified proteins during disease development was further evaluated by bioinformatics analysis using MetaCore™ 6.16 software, which revealed that the majority of these proteins were associated with oncogenesis (reply 1). We found Peroxiredoxin 4 and Gelsolin were linked together, whose levels are influenced as a prognostic and predictive biomarker in CRC tumor tissues with lymph node metastasis (LNM) stage IV. First from this study, interestingly GSN and PRDX4 may determined whether participate in the aggression and progression of CRC cells with the depth of invasion, lymph node metastasis.
- The authors should perform bioinformatics analysis of the 18 proteins detected in order to elucidate their role in patients with lymph node metastasis in colorectal cancer.
Response: Thanks for the comments and have been shown in the reply 1 to perform bioinformatics analysis. Using the Bioinformatics analysis of differentially expressed proteins by MetaCore™ software to understand potential interaction and regulation network is associated with these proteins regarding cellular signaling and gene expression, which was distributed in the extracellular space, membrane and cytoplasm. we found Gelsolin and PRDX4 are linked together, whose levels are influenced as a prognostic and predictive biomarker in CRC tumor tissues with lymph node metastasis (LNM) stage IV.
Reply 2.
Network profile of protein interactions associated with eighteen candidate proteins at the cellular and genetic levels by MetaCore. This network was generated using the shortest path algorithm to map interaction among different proteins. The arrowheads indicated the direction of the interaction. The color of the lines between nodes indicated interactions.
- In Figure 3. Illustrations of hematoxylin and eosin staining and immunohistochemistry of GSN and 352 PRDX4 on CRC tissue microarrays, please include the normal tissues as controls. Also, the legend is confuse, as authors indicate “Intensity”, but in the table they indicated none, low and high as scores. Also, the Node positive and negative were not indicated. Please correct the figure 3 as in Table 3.
Response: We agree with this comment and have performed the hematoxylin and eosin staining to assess the normal tissues as controls in the fig 3. The legend of Figure 4. is Illustrations of hematoxylin and eosin staining and immunohistochemistry of GSN and PRDX4 on intensity scores of CRC tissue microarrays.
Immunostaining was assessed independently by a pathologist in a blinded manner. The staining of Peroxiredoxin 4 and Gelsolin were scored as the product of the staining intensity (on a scale of 0–2: negative = 0, low= 1, high= 2) and the percentage of cells stained (on a scale of 0–3: 0 = zero, 1 = 1%–25%, 2 = 26%–50%, 3 = 51%–100%), resulting in scores on a scale of 0–6 (19). The protocol in previous publication (18, Huang et al., 2018; 19, Pimentel-Nunes et al., 2011) in the page 3, lines 114-117.
- The western blots in Figure 4A display very poor quality. Multiple immunodetected bands are showed. I strongly suggest to improve the western blots for both GSN and PRDX4 proteins in order to conclude.
Response: Thanks for the comments. The Protein lysates of the DLD-1 cells from control scrambled shRNA, PRDX4 shRNA, and GSN shRNA clones were subjected to western blot analysis. Whole-cell lysate proteins of DLD-1 cells were subjected to western blot, with polyclonal antibodies against Peroxiredoxin 4 and Gelsolin, which shown on multiple bands for PRDX4 and GSN inDLD-1 cells. Rabbit polyclonal antibodies (diluted 1:1000) against GSN and PRDX4 were purchased from GeneTex Inc. (GeneTex,TX, USA) (line 126-128).
- Should the authors include immunoblot or qRT-PCR assays to confirm GSN and PRDX4 expression levels in CRC?
Response: We agree with this comment “Another colon cancer cell lines, HCT-116 and SW620 (high metastatic potential, CRC metastasis to LNs and distant organs liver and lungs), continuously by immunoblot were determined for GSN and PRDX4 expression levels. Previous studies showed that HCT116 and SW620 cells are highly metastatic and found that both high levels of CXCR3 and CXCR4 (Int. J. Cancer:132, 276–287 (2013)).
Our data showed that the protein level of Peroxiredoxin 4 and Gelsolin were showed in HCT116 and SW620 cells in reply 3. Further we are planning to study the effects of altered PRDX4 and GSN expression levels on biologic processes relevant to metastasis in human HCT116 and SW620 cells stably expressing by shRNA PRDX4 and shRNA GSN.
Reply 3
- The authors should include in vivo tumor detection assays by PET imaging or zebrafish xenograft models to evaluate the effect of GSN and PRDX4 on metastasis in CRC.
Response: Thanks for the comments. The title in the 3.5. section of manuscript, had showed stably expressing of the DLD-1 cells with GSN shRNA and PRDX4 shRNA led to significantly decreased tumor weight in tumor tissues compared with that of the DLD-1 cells (Control shRNA) in mouse xenografts.
Additional studies are still needed to elucidate the action mechanisms of the GSN and PRDX4 on metastasis in vivo xenograft mouse model, using Bioluminescence imaging of tumor growth and captured by IVIS Spectrum Imaging System.

Reviewer 2 Report
The work, titled "2D- DIGE-MS Proteomics Approaches for Identification of Gelsolin and Peroxiredoxin 4 with Lymph Node Metastasis in Colorectal Cancer," aims to, a combination of fluorescence two-dimensional differential gel electrophoresis (2D-DIGE) and matrix-assisted laser desorption/ionisation time-of-flight mass spectrometry approach was used to search for potential markers for prognosis and intervention of colorectal cancer (CRC) at different stages of lymph node metastasis (LMN). In the end, GSN and RDX4 are novel regulators in CRC lymph node metastasis that may provide new insights into the mechanism of CRC progression and serve as biomarkers for CRC diagnosis at metastatic stage. The work is well written and has the makings of a publication.
Authors should include all information concerning identification of peptides/proteins.
Author Response
Comments and Suggestions for Authors
The work, titled "2D- DIGE-MS Proteomics Approaches for Identification of Gelsolin and Peroxiredoxin 4 with Lymph Node Metastasis in Colorectal Cancer," aims to, a combination of fluorescence two-dimensional differential gel electrophoresis (2D-DIGE) and matrix-assisted laser desorption/ionisation time-of-flight mass spectrometry approach was used to search for potential markers for prognosis and intervention of colorectal cancer (CRC) at different stages of lymph node metastasis (LMN). In the end, GSN and RDX4 are novel regulators in CRC lymph node metastasis that may provide new insights into the mechanism of CRC progression and serve as biomarkers for CRC diagnosis at metastatic stage. The work is well written and has the makings of a publication.
Authors should include all information concerning identification of peptides/proteins.
Response: We appreciate this remark of reviewer #2’s comment and respond to the comment to include all information concerning identification of peptides/proteins. Summary table of mascot protein identification. MALDI-TOF/TOF and Mascot Database searching identified a number of proteins from lymph node metastasis stage I and stage IV of human colorectal cancer.
Protein List: Z:\Hau\20150319_Kuo\1_RE2_01_2231.d\BTDataExchange\ProteinList.WARPResult WARP-LC Method: D:\Methods\WarpLCMethods\Hau_20120417_Human.WarpLCMethod
BioTools Method: hau-201109_human Computer Name: ANALYSIS
|
|
Rel. |
Protein |
Accession |
Score |
MW [kDa] |
pI |
SC [%] |
# Pept. |
RMS [ppm] |
Abs. Inten. |
S/N |
|
1 |
TRUE |
Tropomyosin beta chain OS=Homo sapiens GN=TPM2 PE=1 SV=1 |
TPM2_HUMAN |
304.48 |
32.94461 |
4.509962746 |
25.35211268 |
15 |
197.7672914 |
45964690 |
376.7 |
|
2 |
TRUE |
Translationally-controlled tumor protein OS=Homo sapiens GN=TPT1 PE=1 SV=1 |
TCTP_HUMAN |
125.41 |
19.69662 |
4.689094589 |
30.23255814 |
6 |
393.1462451 |
30126826 |
129.7 |
|
3 |
TRUE |
Proteasome subunit alpha type-5 OS=Homo sapiens GN=PSMA5 PE=1 SV=3 |
PSA5_HUMAN |
71.96 |
26.56527 |
4.589297328 |
21.16182573 |
3 |
57.06291034 |
12389556 |
51.1 |
|
4 |
TRUE |
Glucosidase 2 subunit beta OS=Homo sapiens GN=PRKCSH PE=1 SV=2 |
GLU2B_HUMAN |
75.43 |
60.35719 |
4.177305421 |
3.787878788 |
3 |
254.6466505 |
6284811 |
37 |
|
5 |
TRUE |
Transcription activator BRG1 OS=Homo sapiens GN=SMARCA4 PE=1 SV=2 |
SMCA4_HUMAN |
42.67 |
185.10016 |
8.446624265 |
1.882210079 |
4 |
968.4703684 |
1666514 |
34.6 |
|
6 |
TRUE |
E3 ubiquitin-protein ligase HERC2 OS=Homo sapiens GN=HERC2 PE=1 SV=2 |
HERC2_HUMAN |
41.12332086 |
533.50969 |
5.846865284 |
0.868845676 |
4 |
645.5683854 |
5771713 |
20 |
|
7 |
TRUE |
Tubulin beta-4B chain OS=Homo sapiens GN=TUBB4B PE=1 SV=1 |
TBB4B_HUMAN |
472.47 |
50.25517 |
4.647546299 |
28.98876404 |
18 |
283.5212478 |
101544106 |
357.4 |
|
8 |
TRUE |
N-alpha-acetyltransferase 35, NatC auxiliary subunit OS=Homo sapiens GN=NAA35 PE=1 SV=1 |
NAA35_HUMAN |
51.63444029 |
84.5535 |
6.636065398 |
8.137931034 |
4 |
645.9629643 |
25441331 |
34.9 |
|
9 |
TRUE |
14-3-3 protein zeta/delta OS=Homo sapiens GN=YWHAZ PE=1 SV=1 |
1433Z_HUMAN |
268.5944403 |
27.89879 |
4.574400801 |
48.57142857 |
14 |
174.8582331 |
108082709 |
339 |
|
10 |
TRUE |
Apolipoprotein A-I OS=Homo sapiens GN=APOA1 PE=1 SV=1 |
APOA1_HUMAN |
350.23 |
30.75893 |
5.463584742 |
44.19475655 |
18 |
308.3744777 |
68966977 |
365.9 |
|
11 |
TRUE |
Rho GDP-dissociation inhibitor 1 OS=Homo sapiens GN=ARHGDIA PE=1 SV=3 |
GDIR1_HUMAN |
199.99 |
23.24973 |
4.862644608 |
27.94117647 |
9 |
227.154431 |
75700329 |
262.4 |
|
12 |
TRUE |
ATP synthase subunit d, mitochondrial OS=Homo sapiens GN=ATP5H PE=1 SV=3 |
ATP5H_HUMAN |
72.02 |
18.53652 |
5.06961347 |
41.61490683 |
4 |
65.29820046 |
45461096 |
117.1 |
|
13 |
TRUE |
Elongation factor Tu, mitochondrial OS=Homo sapiens GN=TUFM PE=1 SV=2 |
EFTU_HUMAN |
323.3433209 |
49.85231 |
7.92477118 |
29.20353982 |
15 |
123.131782 |
41297665 |
389 |
|
14 |
TRUE |
Selenium-binding protein 1 OS=Homo sapiens GN=SELENBP1 PE=1 SV=2 |
SBP1_HUMAN |
149.46 |
52.92784 |
5.918708093 |
9.322033898 |
5 |
279.8645111 |
13235837 |
115.6 |
|
15 |
TRUE |
Gelsolin OS=Homo sapiens GN=GSN PE=1 SV=1 |
GELS_HUMAN |
330.6444403 |
86.04334 |
5.858520534 |
23.52941176 |
17 |
322.6283094 |
151223676 |
304.4 |
|
16 |
TRUE |
Peroxiredoxin-4 OS=Homo sapiens GN=PRDX4 PE=1 SV=1 |
PRDX4_HUMAN |
65.88 |
30.7489 |
5.846664536 |
21.40221402 |
4 |
364.6305014 |
23424753 |
124.2 |
|
17 |
TRUE |
Proteasome activator complex subunit 2 OS=Homo sapiens GN=PSME2 PE=1 SV=4 |
PSME2_HUMAN |
186.77 |
27.55538 |
5.446294653 |
33.47280335 |
8 |
113.2878529 |
74450614 |
268.8 |
|
18 |
TRUE |
Heterogeneous nuclear ribonucleoprotein F OS=Homo sapiens GN=HNRNPF PE=1 SV=3 |
HNRPF_HUMAN |
95.31 |
45.98498 |
5.274237773 |
10.60240964 |
4 |
191.8402221 |
24047600 |
64.2 |

Round 2
Reviewer 1 Report
Authors have replied some of the concerns. However, some details remains to be corrected, as no response have been provided to my concerns..
1. Authors should include the graphical abstract and simple summary. No simple summary neither graphical abstract image was included in the revsed version of manuscript.
2. The authors should perform bioinformatics analysis of the 18 proteins detected in order to elucidate their role in patients with lymph node metastasis in colorectal cancer.
Sorry but I cant see the bioifnormatic analysis in the revised manuscript.
3. The western blots in Figure 4A display very poor quality. Multiple immunodetected bands are showed. I strongly suggest to improve the western blots for both GSN and PRDX4 proteins in order to conclude.
Authors did not reply to my concerns, instead of, they describe again the methodology, but not provide the requested western blots.
Author Response
Comments and Suggestions for Authors
Authors have replied some of the concerns. However, some details remains to be corrected, as no response have been provided to my concerns..
- Authors should include the graphical abstract and simple summary. No simple summary neither graphical abstract image was included in the revsed version of manuscript.
Response: We agree with this comment. We have described in the manuscript, page 18, lines 595 to 596, in the fig 11.
- The authors should perform bioinformatics analysis of the 18 proteins detected in order to elucidate their role in patients with lymph node metastasis in colorectal cancer.
Sorry but I cant see the bioifnormatic analysis in the revised manuscript.
Response: We agree with this comment. We have described in the manuscript, page 8, lines 307 to 314, in the fig 3.
- The western blots in Figure 4A display very poor quality. Multiple immunodetected bands are showed. I strongly suggest to improve the western blots for both GSN and PRDX4 proteins in order to conclude.
Authors did not reply to my concerns, instead of, they describe again the methodology, but not provide the requested western blots.
Response: We thanks for this comment to improve the western blots. We have revised in the manuscript, in the fig 5A.

Round 3
Reviewer 1 Report
Authors have replied all my concerns, thus I reccomend to accept for publication the manuscript in its actual form.